# Tabula: Efficiently Computing Nonlinear Activation Functions for Secure Neural Network Inference

**Maximilian Lam**  *maxlam@g.harvard.edu*
*Harvard University*

**Michael Mitzenmacher**  *michaelm@eecs.harvard.edu*
*Harvard University*

**Vijay Janapa Reddi**  *vj@eecs.harvard.edu*
*Harvard University*

**Gu-Yeon Wei**  *guyeon@eecs.harvard.edu*
*Harvard University*

**David Brooks**  *dbrooks@g.harvard.edu*
*Harvard University*

**Reviewed on OpenReview:** *https://openreview.net/forum?id=CXPb4twsrq*

## Abstract

Multiparty computation approaches to secure neural network inference commonly rely on garbled circuits for securely executing nonlinear activation functions. However, garbled circuits require excessive communication between server and client, impose significant storage overheads, and incur large runtime penalties; for example, securely evaluating ResNet-32 using standard approaches requires more than 300MB of communication, over 10s of runtime, and around 5 GB of preprocessing storage. To reduce these costs, we propose an alternative to garbled circuits: Tabula, an algorithm based on secure lookup tables. Our approach precomputes lookup tables during an offline phase that contains the result of all possible nonlinear function calls. Because these tables incur exponential storage costs in the number of operands and the precision of the input values, we use quantization to reduce these storage costs to make this approach practical. This enables an online phase where securely computing the result of a nonlinear function requires just a single round of communication, with communication cost equal to twice the number of bits of the input to the nonlinear function. In practice our approach costs 2 bytes of communication per nonlinear function call in the online phase. Compared to garbled circuits with 8-bit quantized inputs, when computing individual nonlinear functions during the online phase, experiments show Tabula with 8-bit activations uses between 280-560× less communication, is over 100× faster, and uses a comparable (within a factor of 2) amount of storage; compared against other state-of-the-art protocols Tabula achieves greater than 40× communication reduction. This leads to significant performance gains over garbled circuits with quantized inputs during the online phase of secure inference of neural networks: Tabula reduces end-to-end inference communication by up to 9× and achieves an end-to-end inference speedup of up to 50×, while imposing comparable storage and offline preprocessing costs.

## 1 Introduction

Secure neural network inference seeks to allow a server to perform neural network inference on a client's inputs while minimizing the data leakage between the two parties. Concretely, the server holds a neural

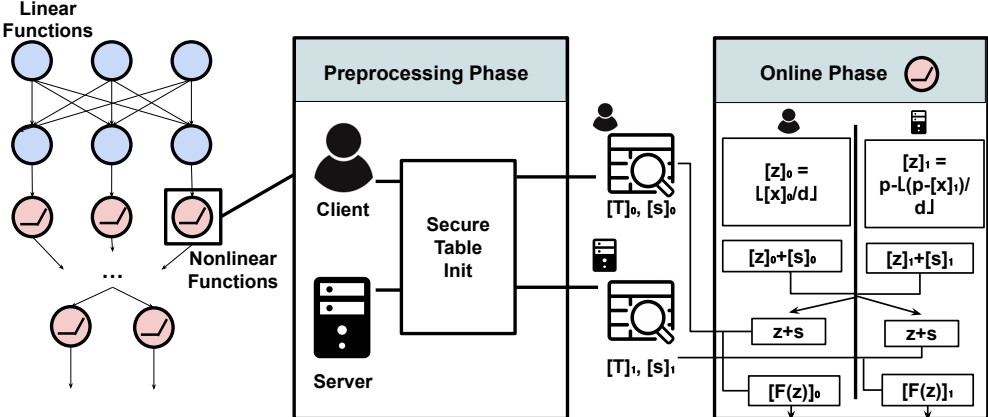

Figure 1: The TABULA approach to computing nonlinear functions for secure neural network inference. TABULA precomputes lookup tables $[T]_0$, $[T]_1$ stored on client and server respectively, and also initializes shares of the secret $s$ so that the client holds $[s]_0$ and the server holds $[s]_1$. The lookup tables $[T]_i$ contain the result of all possible nonlinear function calls to an activation function and uses quantization to make storing all possible function calls in the table feasible. These lookup tables map secret shares of the quantized inputs to the nonlinear function to secret shares of the output of the activation function. During the online phase, these lookup tables enable extremely efficient nonlinear activation function execution and proceeds by 1) securely truncating the inputs, 2) reconstructing a blinded index and 3) looking up the blinded index in the lookup tables $[T]_i$. Our code is released at `https://github.com/tabulainference/tabula`.

network model $M$ while the client holds an input $x$. The objective of a secure inference protocol is for the client to compute $M(x)$ without revealing any additional information about the client's input $x$ to the server, and without revealing any information about the server's model $M$ to the client. A protocol for secure neural network inference brings significant value to both the server and the clients. The clients' sensitive input data is kept secret from the server, shielding the user from malicious data collection. Additionally, the client does not learn anything about the server's model, which prevents the model from being stolen by competitors.

Current state-of-the-art multiparty computation approaches to secure neural network inference require significant communication between client and server, lead to excessive runtime slowdowns, and incur large storage penalties (Mishra et al., 2020a; Ghodsi et al., 2021; Jha et al., 2021; Rathee et al., 2020; Juvekar et al., 2018; Cho et al., 2021). The source of these expenses is computing nonlinear activation functions with garbled circuits (Yao, 1986). Garbled circuits are costly in terms of computation, communication, and storage. Concretely, executing ReLU activation functions using garbled circuits requires over 2 KB of communication per scalar element of the input (Mishra et al., 2020a) and imposes over 17 KB of preprocessing storage per scalar element of the input (Mishra et al., 2020a; Ghodsi et al., 2021). These costs make state-of-the-art neural network models prohibitively expensive to deploy: on ResNet-32, state-of-the-art multiparty computation approaches for a single secure inference require more than 300 MB of data communication (Mishra et al., 2020a), take more than 10 seconds for an individual inference (Mishra et al., 2020a), and impose over 5 GB of preprocessing storage per inference (Ghodsi et al., 2021). These communication, runtime, and storage costs pose a significant barrier to deployment, as they degrade user experience, drain clients' batteries, induce high network expenses, and eliminate applications that require sustained real time inference.

To replace garbled circuits and other methods (Rathee et al., 2020; Huang et al., 2022) for privately computing nonlinear functions, we propose TABULA, a two-party secure protocol to efficiently evaluate neural network nonlinear activation functions. During an offline preprocessing phase, TABULA generates tables that contain the encrypted result of evaluating a nonlinear activation function over a range of all possible quantized inputs. New tables are precomputed for each nonlinear function performed during inference, and these tables are split across client and server. Then, at inference time, TABULA performs two steps to evaluate a nonlinear activation function: 1) securely quantize neural network activation inputs down to the precision of the range

of the inputs to the precomputed tables and 2) securely lookup the result of the activation function using a two-party secure table lookup procedure (Ishai et al.; Keller et al., 2017; Damgård & Zakarias, 2016). By heavily quantizing neural network activations and reducing the space of inputs to the nonlinear activation function, TABULA enables storing all possible results of the activation function in a table without requiring an infeasibly large amount of memory. This allows the application of the subsequent two-party secure table lookup protocol, which is efficient and has low storage, communication, and computation overhead.

TABULA achieves significant improvements over garbled circuits and other (Rathee et al., 2020; Huang et al., 2022) approaches for securely computing nonlinear functions on important system metrics such as communication and runtime, while maintaining or even improving storage costs.

- **Runtime**
  TABULA offers significant runtime improvements over garbled circuits with quantized inputs due to the simplicity of the online phase of the secure table lookup protocol (Keller et al., 2017; Ishai et al.; Damgård & Zakarias, 2016). TABULA's runtime for an individual activation function is the cost of transferring a single secretly shared value between parties, and performing a single memory access on the subsequent value. Our results show that when computing individual functions, TABULA is over $100\times$ faster than garbled circuits with quantized inputs. This leads to significant overall runtime improvements when performing secure neural network inference. Our results show that across various standard networks (LeNet, ResNet-32, ResNet-34, VGG) TABULA achieves up to $50\times$ runtime speedup compared to garbled circuits with quantized inputs. Additionally, during the online phase, the TABULA protocol requires just one table lookup; this is significantly less computation compared to schemes that use function secret sharing (FSS), which require computing PRGs like AES-128 (Wagh, 2022; Gupta et al., 2022; Agarwal et al., 2022; Boyle et al., 2019; 2020; Ryffel et al., 2021).

- **Communication**
  TABULA requires significantly less communication than garbled circuits with quantized inputs and also significantly less communication versus other state-of-the-art approaches like (Rathee et al., 2020). TABULA's communication cost for a single activation function is the cost of communicating a single secretly shared element between parties, and is independent of the complexity of the nonlinear function. Our experiments show that, compared to garbled circuits with quantized inputs, communication required for a single nonlinear function call is reduced by a factor of over $280 - 560\times$ leading to an overall communication reduction of up to $9\times$ on various standard neural networks. Additionally, compared to other state-of-the-art protocols for computing nonlinear functions like (Rathee et al., 2020), we show TABULA reduces communication by up to $40\times$ on a per-operation basis, leading to up to $10\times$ reduction in communication when performing end-to-end private inference on various neural networks.

- **Storage and Memory**
  TABULA utilizes comparable storage and memory as garbled circuits with quantized inputs. TABULA's table sizes are dictated by how heavily quantized the activations are and increase exponentially with the precision of the activations. Notably, TABULA's table sizes affect the precision of the activation function and hence affect neural network accuracy. However, as neural network activations may be significantly quantized without significantly affecting neural network quality (Ni et al., 2020; de Bruin et al., 2020; Zhao et al., 2020; McKinstry et al., 2019), the sizes of these individual tables may be reduced enough to allow a comparable or smaller amount of storage than garbled circuits with quantized inputs. Generally, across different models, TABULA uses between $.25 - 2\times$ as much memory as garbled circuits while maintaining similar model quality. Like garbled circuits, TABULA requires a new table for each individual nonlinear operation to maintain security.

A comparison of our work against others across some of these axes is shown in Table 1.

| | Works | Comm. Cost (per-op) | Runtime Cost (per-op) | Preproc. Storage Cost (per-op) | Supported Nonlinear Operations | Mode of Operation | Online Phase Computation |
|---|---|---|---|---|---|---|---|
| Tabula | Ours | 2B (Any function, must fit in lookup table) | .55 us | 16KB | Any | Secure Lookup Tables | 1 RAM lookup |
| Garbled Circuits | Delphi (Mishra et al., 2020a), Gazelle (Juvekar et al., 2018), SecureNN(Mohassel & Zhang, 2017), DeepReduce (Jha et al., 2021), Circa (Ghodsi et al., 2021) | 562B | 70 us | 17.5KB | Any | GCs (Yao, 1986) | - |
| Tree-Based Comparator | CryptFlow2 (Rathee et al., 2020), Cheetah (Huang et al., 2022) | 96B | - | - | ReLU only | Oblivious Transfer (Lo, 1997) | - |
| Function Secret Sharing (FSS) | Pika (Wagh, 2022), Llama (Gupta et al., 2022), Agarwal et al. (2022) Ariann(Ryffel et al., 2021) | 2B (8-bit compare, Boyle et al. (2019)) | - | 512B (Boyle et al., 2019) | Any | FSS (Boyle et al., 2019) (Boyle et al., 2020) | PRG (i.e: AES-128) |

Table 1: Comparison of our work against other approaches for securely computing nonlinear activation functions across selected axes. Unless specified costs refer to the cost of the online phase. Compared to garbled circuits, the most widely used state-of-the-art protocol for securely computing nonlinear functions, our approach sees significant improvements in communication and runtime at comparable storage costs. We also compare our approach against less generic protocols for non-linear function computation (tree-based comparator, limited to only ReLU) on the basis of communication where we again see considerable improvements. Finally, compared to function secret sharing (FSS) schemes, our approach is comparable in attaining low communication cost while being computationally more efficient.

## 2 Related Work

### 2.1 Multiparty Computation Approaches to Secure Neural Network Inference

Multiparty computation approaches to secure neural network inference have been limited by the costs of computing both the linear and nonlinear portions of the network (Mohassel & Zhang, 2017; Rouhani et al., 2017; Rathee et al., 2020; Keller, 2020). Recent works like Minionn, Gazelle and Delphi (Liu et al., 2017; Juvekar et al., 2018; Mishra et al., 2020a; Lehmkuhl et al., 2021; Rathee et al., 2020; Jha et al., 2021; Cho et al., 2021; Ghodsi et al., 2021) have optimized the linear operations of secure neural network inference via techniques like preprocessing to the point they are no longer a major system bottleneck (Mishra et al., 2020a). Hence, current state-of-the-art approaches to secure inference like Minionn, Gazelle, Delphi, and CrypTFlow2 are primarily bottlenecked by nonlinear operations. Specifically, these approaches rely on garbled circuits, or a circuit-based protocol, to compute nonlinear activation functions (e.g: ReLU) (Liu et al., 2017; Juvekar et al., 2018; Mishra et al., 2020a; Keller & Sun, 2021; Dalskov et al., 2020), resulting in notable drawbacks including high communication, runtime and storage costs.

Our approach addresses the problems posed by garbled circuits by eliminating them altogether. Our method is centered around precomputing lookup tables containing the encrypted results of nonlinear activation functions, and using quantization to reduce the size of these tables to make them practical.

### 2.2 Lookup Tables for Secure Computation

Lookup tables have been used to speed up computation for applications in both secure multiparty computation (Launchbury et al., 2012; Damgård et al., 2017; Keller et al., 2017; Rass et al., 2015; Dessouky et al., 2017) and homomorphic encryption (Li et al., 2019; Crawford et al., 2018). These works have demonstrated that lookup tables may be used for garbled circuits computation Heath et al. (2024) or as an efficient alternative to garbled circuits, provided that the input space is small. Prior works have primarily focused on using lookup tables to speed up traditional applications like computing AES (Keller et al., 2017; Damgård et al., 2017; Launchbury et al., 2012; Dessouky et al., 2017) and data aggregation (Rass et al., 2015). Notable exceptions include (Crawford et al., 2018) which focuses on linear regression, and (Rathee et al., 2021) which applies variants of a lookup table as part of the protocol to secure machine learning inference. Lookup tables are

also widely used as cryptographic primitives in SNARKS (Setty et al., 2023; Arun et al., 2023), notably as efficient primitives for non-arithmetic operations inside circuits (Arun et al., 2023).

To the best of our knowledge, there exists little prior work which applies secure lookup tables to the private execution of large neural networks. Most current state-of-the-art secure inference systems like (Mishra et al., 2020a; Mohassel & Zhang, 2017; Ghodsi et al., 2021; Jha et al., 2021) use garbled circuits. Two exceptions to this include (Rathee et al., 2020; Huang et al., 2022), which use a tree-based secure comparator. However, the tree-based secure comparator used in (Rathee et al., 2020; Huang et al., 2022) is significantly limited to only the ReLU activation function, and still requires significant computation and communication overhead. Another work, (Rathee et al., 2021), does indeed use lookup tables as part of their protocol for evaluating activation functions, but crucially focuses on ensuring numerical precision, leading to lower system performance. We highlight that a key distinction in our use of lookup tables is that we store all possible results of the nonlinear function in these tables, which uses exponential storage. This storage is made manageable by heavily quantizing the neural network. This strategy of securely evaluating a function through lookup tables, although known theoretically (Ishai et al.; Damgård & Zakarias, 2016), to the best of our knowledge has not been applied practically until now, due to the exponential storage costs. The secure lookup table approach of "storing everything in a table" is remarkably well suited to securely and efficiently computing neural network nonlinear activation functions for two reasons: 1) neural network activations may be quantized to extremely low precision with little degradation to accuracy and 2) neural network activation functions are single operand. These two factors allow us to limit the size of the lookup table to be sufficiently small to be practical, and consequently we can achieve the significant performance benefits of secure lookup tables at runtime (i.e two orders of magnitude less communication). While work on quantization applied to secure inference exists (Dalskov et al., 2020; Keller & Sun, 2021), these works do not combine this property with lookup tables for evaluating nonlinear functions. In summary, we emphasize that prior works that use secure lookup tables have either applied them towards non-ML applications (i.e: MPC for AES-128), or have not leveraged exponential-sized secure lookup tables in combination with neural network quantization to make them practical; although the exponential-sized secure lookup table approach for securely computing functions is known, the unique combination of this technique with neural network quantization has not been previously explored. In this work, we demonstrate that this unique combination of techniques can be applied to dramatically reduce the costs of secure neural network inference.

## 2.3 Function Secret Sharing

The secure lookup table approach (Ishai et al.) employed by TABULA is related to function secret sharing (FSS) approaches used in various private neural network inference approaches like Wagh (2022); Gupta et al. (2022); Agarwal et al. (2022). The secure lookup table approach of "storing all function inputs/outputs in a secure lookup table" can be theoretically categorized as a FSS approach. But there are several concrete differences between the secure table lookup approach TABULA uses compared to traditional FSS approaches. These distinctions lead to significant runtime differences. Concretely, our secure lookup table approach incurs exponential storage costs which necessitates aggressive activation quantization to make storage costs practical. However, this approach also enables a highly efficient online phase which requires just one 8-bit memory access (in addition to the 2B communication between parties). FSS approaches, on the other hand, rely on using distributed point functions (DPFs) or distributed comparison functions (DCFs), (Boyle et al., 2019; 2020) which in turn require evaluating PRGs (i.e: AES-128). Specifically, a table lookup using FSS requires at least $\log(N)$ PRG or AES-128 evaluations, where $N$ is the number of entries in the table, leading to 8-16 AES-128 evaluations per activation function call. This cost increases for more complex nonlinear functions (Boyle et al., 2019; 2020). Evaluating PRGs like AES-128 is comparatively more expensive than TABULA which requires just 1 8-bit memory access, as a modern processor even with hardware acceleration computes only around 100M AES-128 operations (aes), whereas a modern processor has a memory bandwidth in the 100GB/s range. As such, TABULA is much more computationally efficient compared to FSS schemes, though as a drawback requires aggressive quantization to make practical. Another benefit to TABULA is that its communication cost is always 2B regardless of the nonlinear function being securely computed. This is not the case for function secret sharing where more complicated nonlinear functions may cost more than 2B Boyle et al. (2019; 2020). Furthermore, a third advantage is that TABULA exhibits information theoretic security in the online phase, while function secret sharing schemes are only computationally secure up to a factor of the

security parameter (Boyle et al., 2020); that is, the protocol leaks no information about the underlying data unlike FSS schemes, as FSS schemes rely on pseudorandom functions. A final and notable advantage is that TABULA is much simpler than FSS schemes that rely on computing distributed point functions or distributed comparison functions, which are complex cryptographic primitives. TABULA's main observation is that neural network activations can be aggressively quantized to 8-bits and below with acceptable accuracy degradation and thus enable the use of exponential-sized lookup tables, thereby avoiding the need for evaluating PRGs like AES-128 during online inference.

## 3  Tabula: Efficient Nonlinear Activation Functions for Secure Neural Network Inference

### 3.1  Background

**Secure Inference Objectives, Threat Model**
Secure neural network inference seeks to compute a sequence of linear and nonlinear operations parameterized by the server's model over a client's input while revealing as little information to either party beyond the model's final prediction. Formally, given the server's model's weights $W_i \in \mathbb{F}_p^{M_i \times L_i}$ and the client's private input $x$, the goal of secure neural network inference is to compute $a_i = A(W_i a_{i-1})$ where $a_0 = x$ and $A$ is a nonlinear activation function, typically ReLU. $W_i \in \mathbb{F}_p^{M_i \times L_i}$ are the weights of the neural network represented as a fixed point number in the finite field of modulus $p$. The dimensions $M_i$ and $L_i$ correspond to the output and input dimensions to the linear layer at $i$. Convolutions may be cast as a matrix multiply and fit within this framework.

State-of-the-art secure neural network inference protocols like Delphi operate under a two-party semi-honest setting (Mishra et al., 2020a; Lehmkuhl et al., 2021), where only one of the parties is corrupted and the corrupted party follows the protocol. Importantly, these secure inference protocols do not protect the architecture of the neural network being executed, only its weights, and furthermore do not secure any information leaked by the predictions themselves (Mishra et al., 2020a). As we follow Delphi's protocol for the overall secure execution of the neural network these security assumptions are implicitly assumed.

**Cryptographic Primitives, Notations, Definitions**
TABULA utilizes standard tools in secure multiparty computation. Our protocols operate over additively shared secrets in finite fields. We denote $\mathbb{F}_p$ as a finite field over $n$-bit prime $p$. We use $[x]$ to denote a two party additive secret sharing of the scalar $x \in \mathbb{F}_p$ such that $x = [x]_0 + [x]_1$, where party $i$ holds additive share $[x]_i$ but knows no information about the other share.

**Delphi Secure Inference Protocol**
The Delphi framework is a set of protocols consisting of an offline preprocessing phase and an online secure inference phase for securely evaluating neural networks over private client data without revealing to the client the weights of the neural network. The Delphi framework is concerned with the overall evaluation of the neural network (both linear and nonlinear layers). TABULA fits into the Delphi framework by replacing their use of garbled-circuits protocols for secure nonlinear function evaluation, which is the most computationally expensive part of their protocol Mishra et al. (2020a). To understand how TABULA fits into the Delphi framework (Mishra et al., 2020b), we outline how this protocol operates.

- **Per-Input Preprocessing Phase**
  This phase prepares for the secure execution of a single input. The purpose of the preprocessing phase is to initialize the parties with correlated randomness that enables efficient online inference. This phase, as specified in the Delphi paper, requires the use of linearly homomorphic encryption (Mishra et al., 2020a) to ensure that the parties do not reveal to each other their secret blinding factors which would compromise the privacy of the entire protocol. For each linear layer $W_i \in \mathbb{F}_p^{M_i \times L_i}$, the client generates a random vector $R_c \in \mathbb{F}_p^{L_i}$ where $L_i$ is the length of the inputs to the current linear layer. The client encrypts $R_c$ with their linearly homomorphic public encryption key $k$ to $Enc_k(R_c)$ and sends this value to the server. The server, upon receiving $Enc_k(R_c)$, generates their own secret vector $R_s \in \mathbb{F}_p^{M_i}$ where $M_i$ is the length of the outputs of the current linear layer. The server then encrypts

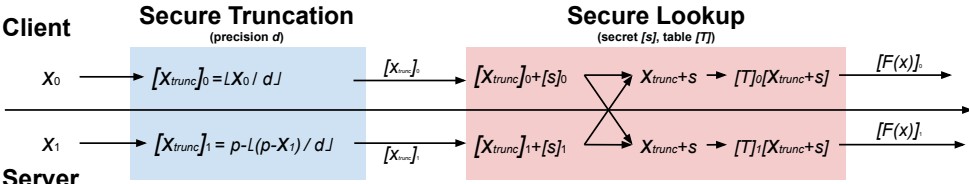

Figure 2: TABULA online protocol. Initially, the client and server hold secret shares of the input $[x]$. Both parties begin by executing the secure truncation protocol to obtain shares of $[x_{trunc}]$. Then, the client and server perform the secure table lookup protocol, where they exchange blinded secrets $[x_{trunc}]_i + s_i$ to compute $x_{trunc} + s$. Finally, they use this value as an index into local lookup tables to compute $T_i[x_{trunc} + s]$ which are secret shares of the result of the nonlinear function evaluation.

$R_s$ with the client's public key $k$ to obtain $Enc_k(R_s)$. The server then computes and returns to the client $Enc_k(W_i R_c + R_s)$. The client decrypts this value to obtain $W_i R_c + R_s$ which is then stored in preparation for the online inference phase.

- **Online Inference Phase**
  This phase performs inference on the client's input. For linear layers, the client and server begin with additive secret shares of the linear layer's input $x$. That is the client and server hold $[x]_0$ and $[x]_1$ respectively, such that $[x]_0 + [x]_1 = x$. As the initial step, the client adds $[x]_0$ with that layer's $R_c$ to obtain $[x]_0 + R_c$. Then the client sends this vector to the server who adds their own share of the layer input $[x]_1$ to obtain $x + R_c$. The server, upon calculating $x + R_c$, then computes $W_i(x + R_c) + R_s = W_i x + W_i R_c + R_s$ (recall that $R_s$ was the secret vector that the server generated for this particular layer). At this point, the client holds $W_i R_c + R_s$ from the preprocessing phase and the server has computed $W_i x + W_i R_c + R_s$. The difference between these two values is $W_i x$. Thus the two parties have obtained a secret sharing of $W_i x$. Then, the two parties must compute a nonlinear activation function over these shares to obtain shares of the activations; in the Delphi framework, garbled circuits are employed to securely perform this operation Mishra et al. (2020a), and it is by replacing this part of the protocol that TABULA obtains considerable computational gains. After calculating shares of the activations, the secure inference phase repeats starting from the linear part of the protocol for the rest of the layers of the network.

As stated, after performing the protocol for the linear phase, the client and server hold secret shares of the input to the nonlinear activation function $F$. Hence we need to construct a secure protocol for performing nonlinear activation functions. This protocol must operate such that the client and server, each holding a secret share of $x$, can calculate secret shares of $F(x)$ without leaking any information about $x$ itself. $F$, the nonlinear function for neural network activations, is typically ReLU, but may also be include other nonlinear functions like sigmoid or tanh. As a note, we emphasize that details on the homomorphic preprocessing phase can be found in the Delphi paper (Mishra et al., 2020a).

## 3.2 Tabula for Securely and Efficiently Evaluating Neural Network Nonlinear Activation Functions

TABULA is divided into a preprocessing phase that initializes a lookup table for each individual nonlinear function call used in the neural network, and an online phase which securely quantizes the activation inputs and looks up the result of the function in the previously initialized tables. An overall figure for our protocol is shown in Figure 1. We emphasize that our paper primarily focuses on the online phase of execution, which determines the system's real time response speed after knowing a client's input, rather than the preprocessing phase, which may be done offline without knowing the client's input data. We also develop a secure and reasonably efficient algorithm for the preprocessing phase of Tabula and conduct thorough experiments in the results section to demonstrate its viability and effectiveness. We leave further innovations to the preprocessing algorithm to future research.

Below we describe the core building blocks that TABULA utilizes, namely, the secure lookup table procedure (Ishai et al.; Keller et al., 2017) and secure truncation protocol (Mohassel & Zhang, 2017). We then describe TABULA's online and preprocessing execution phase, and detail its security, communication, and storage properties.

**Secure Lookup Table Procedure**

We employ the concepts of (Ishai et al.) to enable the computation of a nonlinear function call through a table lookup. By using an exponential amount of preprocessing storage, we obtain a secure protocol under the semi-honest threat model where communication complexity depends only on the size of its input operands, regardless of the complexity of the function being computed. Concretely, we precompute all possible results of a nonlinear function and store them in a table, and utilize these secure tables during the online phase of secure inference. Computing nonlinear functions in the clear is extremely efficient computationally (i.e: cleartext comparison operations), and so the bulk of the pre-computation workload is spent on MPC operations (i.e: Beaver triple multiplication). The lookup table approach is similar to that described in (Keller et al., 2017; Ishai et al.). Like in garbled circuits, TABULA requires new circuits per operation to maintain security.

Given a table $T[x] = F(x)$, where $F : \mathbb{F}_p \to \mathbb{F}_p$ is the target nonlinear function operating over scalars, we initialize a shared table $[T]$ across the parties, so the client holds $[T]_0 \in \mathbb{F}_p^p$ and server holds $[T]_1 \in \mathbb{F}_p^p$. A secret scalar $s \in \mathbb{F}^p$ unknown to both parties is generated and shared between the client and the server, with the client and server holding $[s]_0$ and $[s]_1$ respectively. The shared table $T$ is constructed such that $[T][s + x] = [F(x)]$ for all values of $x \in \mathbb{F}_p$ for some modulus $p$ that determines the precision to compute the nonlinear function. Concretely, this means two tables are generated, $[T]_0$ and $[T]_1$ such that $[T]_0[s + x] + [T]_1[s + x] = F(x)$; both client and server coordinate to initialize their local $[T]_i$ in an offline preprocessing phase. The online phase, given such a shared table, is then straightforward. Initially, the client holds $x_0$ and the server holds $x_1$. The client sends to the server $x_0 + s_0$ while the server sends to the client $x_1 + s_1$. This allows both parties to obtain the true value of $x + s$. Both parties then look up this value in their corresponding tables: client looks up $[T]_0[x + s]$, server looks up $[T]_1[x + s]$, the sum of which is $F(x)$. Security is maintained in the online phase as a new table $[T]$ and secret $s$ are used per function call, with $s$ being unknown to either party, perfectly blinding the secret value $x$.

Securely initializing shared table $[T]$ from $T$ in the offline preprocessing phase is based on the fact that given an index into a table, a table lookup can be cast as a dot product between the entire table with an indicator vector containing a one in the position of the table index (e.g: the one hot vector encoding of the index). Subsequently, a secure two party demux procedure (Keller et al., 2017) transforms secret shares of $[s]$ into secret shared vectors $[s']$ which sum to an indicator vector with a one at the $s$'th position. Finally, a dot product for each entry of the table can be performed to compute $[T]$: $T[x] \times [s'_0] + T[x+1] \times [s'_1] + ... + T[x+n] \times [s'_n] = [T[s+x]]$. We develop an efficient and secure protocol for initializing TABULA tables based on these concepts later in the text.

**Secure Truncation**

As the size of $[T]$ increases linearly with the size of the field $\mathbb{F}_p$ it becomes necessary to truncate or quantize $[x]$ to prevent $[T]$ from being impracticably large. Linear layers are required to use larger finite fields to ensure that their dot products are computed correctly without overflow. Thus, the input to TABULA is a value secret shared in a larger field, and a secure truncation method is required to switch to a smaller field so that the encoded value may be used to index into a feasibly sized table.

We use the secure truncation method in (Mohassel & Zhang, 2017) to achieve this. Given a truncation factor $d$ which specifies the precision of the activation inputs, the client and server perform the secure truncation protocol: the client computes $\lfloor [x]_0/d \rfloor$ and the server computes $p - \lfloor (p - [x]_1)/d \rfloor$. After the truncation protocol is performed, the resulting expressions the client and server hold sum to either $[\lfloor [x]/d \rfloor + 1]$ or $[\lfloor [x]/d \rfloor]$ with probability proportional to $1 - \frac{k}{p}$ where $k$ is the maximum value $x$ may take, and $p$ is the maximum value of the finite field of the previous linear layer (Mohassel & Zhang, 2017).

These small off by one errors, like quantization error, have little impact on model quality due to neural networks' resilience to noise. However, with probability proportional to $\frac{k}{p}$, a large error occurs that is pessimistically assumed to ruin correctness. To reduce the probability of these catastrophic errors, it is necessary to use a large finite field modulus for linear layers.

In practice, we use a 64-bit finite field modulus to reduce the chance that a secure truncation operation catastrophically fails to less than $\frac{1000}{2^{64}}$. Hence, by configuring the modulus appropriately, with high probability, the secure truncation protocol computes the correctly truncated value with a small off by one error which may be tolerated by neural networks (Ni et al., 2020; Reagan et al., 2018).

Requiring 64 bits for the field increases the communication cost required by the linear portions of the protocol over other approaches that commonly use 32 bits, however, the reduction in communication cost by using TABULA tables more than makes up for this communication penalty, and this is shown in the results (note without TABULA we use 32-bit precision for the linear layers).

In our experiments, using a 64-bit field size was essential to maintaining accuracy; using a 32-bit field size TABULA saw considerably worse accuracy due to catastrophic failures from the secure truncation protocol, as a single catastrophic failure anywhere in the computation propagates throughout the network and ruins the entire inference. We refer to (Mohassel & Zhang, 2017) for more details. Developing more effective secure truncation techniques is an important topic for further research.

We emphasize that there is a distinction between the field size used for the linear layers and the number of bits for the activations. Concretely, the 64-bit field sizes are used to represent the fixed-point values that are operands to the linear layers of the network, allowing for near full precision multiply-accumulates during linear operations. This is opposed to quantized $n$-bit (generally $n$=8) inputs which are used as inputs to the lookup table, which determines the amount of memory used for the tables, as each table has $2^n$ entries. Our method is functionally equivalent to performing the linear operation over 64-bit fixed point values, quantizing the result down to $n$ bits, computing the activation function over the truncated result, then scaling back up to 64-bit fixed point values for the next linear operation.

**Tabula Online Phase**

Given these fundamental building blocks, we describe the TABULA protocol. In the preprocessing phase, TABULA generates multiple shared tables $[T]$ as described above for each nonlinear function call that is performed when executing the neural network. How much to truncate/quantize the network' activations is chosen offline to maximize network accuracy. In the online execution phase, TABULA quantizes the inputs to the activation function and uses this input to securely lookup up the result of the function. The full protocol is shown in Figure 2. The security of TABULA is ensured by the security of the secure truncation protocol (Mohassel & Zhang, 2017) and the secure table lookup protocol (Keller et al., 2017; Damgård & Zakarias, 2016; Ishai et al.).

**Tabula Online Phase Communication and Storage Cost**

TABULA achieves significant communication benefits during the online phase at comparable storage costs. As shown in Figure 2, TABULA requires just one round of communication to compute any arbitrary function, unlike garbled circuits, which may require multiple rounds for more complex functions. As an example, ReLU implemented using garbled circuits takes two rounds, whereas TABULA requires just one. Additionally, communication complexity is independent of the complexity of the nonlinear function being computed. Specifically, revealing $s_x$ requires both parties to send their local shares, each nonlinear activation call incurs communication cost corresponding to the number of bits in $F_p$. Since we use 64-bit $F_p$, this results in 16 bytes of communication per activation function, the cost of transferring 8-byte field values back and forth. However, we can apply an optimization to reduce this down to twice the cost of transferring the *size of the input to the table*, rather than the field size. If the size of the table is $2^b$ entries, and if finite field size $p$ is also power of two, then we can have party $i$ first mod their secret shares by $2^b$ before exchanging them. Hence, the two parties hold $[x_{trunc}]_i \bmod 2^b$ before adding their secret shares of the table secret $[s]_i$ to the value and exchanging it; this brings down the total cost of the protocol to $2 \times b$ bits. Modding by $2^b$ yields the correct answer as $x_{trunc} \bmod p = [x_{trunc}]_0 + [x_{trunc}]_1 + pl$ for some $l$, and then $(x_{trunc} \bmod p) \bmod 2^b = ([x_{trunc}]_0 + [x_{trunc}]_1 + pl) \bmod 2^b = ([x_{trunc}]_0 + [x_{trunc}]_1) \bmod 2^b = ([x_{trunc}]_0 \bmod 2^b) + ([x_{trunc}]_1 \bmod 2^b)$. This equivalence shows that the two parties can first perform a modulus of their shares with $2^b$, and that their shares would still sum up to the original sum with the correct modulus. With this optimization, communication is now $2 \times b$ bits per nonlinear function call; if we use 8-bit activations, then $b = 8$, and we use 2 bytes of communication total per call during the online phase, an $8\times$ improvement over the 16 bytes as previously stated.

**Generate Indicator Vector & Share**

**Beaver Triple Outer Product**

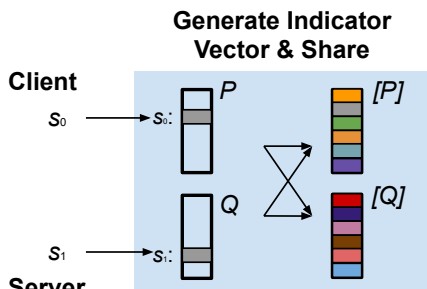
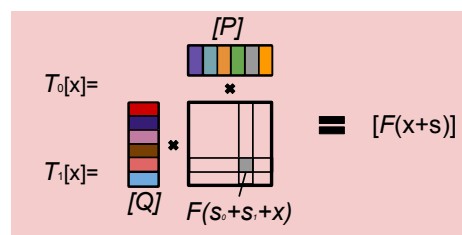

Figure 3: TABULA preprocessing protocol. Client and server generate secrets $s_0$, $s_1$ and encode them in an indicator vector (i.e: construct a vector of length equal to the field size, then setting a one to the position of the party's secret index). The parties then secret share this indicator vector with the other party. To obtain the entry for $T_i[x]$, the parties compute an outer product between the shared indicator vectors and a 2-dimensional table containing $F(m + n + x)$ (where $m, n$ span the two dimensions of the table), which obtains $[F(x + s)]$ where $s = s_0 + s_1$. This works as the 2-D coordinates formed by where the indicator vectors are set privately select $m, n$ through the dot-product; since this is done via private MPC operations, no information is leaked to either party about their corresponding secrets.

Storage and memory, as mentioned previously, grow exponentially with the precision that is used for activations and linearly with the number of activations in the neural network. Storage costs are thus $n \times 2^k \times N_a$ bits where $n$ is the number of bits to use for the the output of the activation function, $k$ is the number of bits to the input of the activation function, and $N_a$ is the total number of activations that are performed by the neural network. The majority of the storage cost comes from the $2^k$ factor, the size of the individual tables, which grows exponentially with input space / precision of the activations. However, as neural network activations may be heavily quantized down without significantly affecting model quality (Ni et al., 2020; de Bruin et al., 2020; Zhao et al., 2020), we can reduce this factor enough to be practical; we also highlight more advanced techniques like using a variable number of bits per layer of the network can be employed for better performance (Dong et al., 2019). We verify that quantization has negligible impact on model quality in our results. We highlight that every bit of precision that is trimmed from the activation yields a factor of two reduction in storage and memory costs, and hence more advanced quantization techniques (Ni et al., 2020; de Bruin et al., 2020; Zhao et al., 2020) to reduce precision yields significant benefits. As storage and memory varies with the precision of activations that is used, there is a natural tradeoff between the accuracy of the model and the achieved memory/storage requirement. We examine these tradeoffs in the results.

### 3.3 Tabula Preprocessing Phase

Similar to garbled cicuits, TABULA tables require a preprocessing phase that initializes the client and server with a single-use table that is used once per activation function call during the inference phase. We develop a secure and efficient protocol for initializing TABULA tables, detailed below.

**Preprocessing Phase Problem Statement**
Given a nonlinear function $F : \mathbb{F}_p \to \mathbb{F}_p$, we wish to securely initialize the client and server with tables that map any possible input in $\mathbb{F}_p$ to secret shares of the result of the nonlinear function $F$. Specifically, we wish to initialize on the client a table $[T]_0 \in \mathbb{F}_p^p$, and on the server a table $[T]_1 \in \mathbb{F}_p^p$, such that $[T]_0[s + x] + [T]_1[s + x] = F(x)$, where $s \in \mathbb{F}_p$ is a secret unknown to both client and server. Additionally, at the end of the protocol, we want the client to hold $s_1 \in \mathbb{F}_p$ and server to hold $s_2 \in \mathbb{F}_p$ such that $s_1 + s_2 = s$.

**Tabula Secure Preprocessing Protocol**
To achieve this preprocessing step securely, the client and server first randomly generate $s_0$ and $s_1$ respectively, and $s$ is implicitly defined as $s_0 + s_1$ (though, as the parties do not know each others' secrets, they hence do not

know what $s$ is). Then, the client generates a random indicator vector $P \in \mathbb{F}_p^p$ such that $P[x] = \begin{cases} 1 & x = s_0 \\ 0 & x \neq s_0 \end{cases}$;

the server similarly initializes $Q \in \mathbb{F}_p^p$ with $s_1$. Client and server exchange shares of $P$ and $Q$ respectively, hence, both parties hold secret shares $[P]$ and $[Q]$ while leaking no information about $s_0$ or $s_1$ to either party. Finally, client and server jointly initialize their table $T_i[x] = \sum_{m=0}^p \sum_{n=0}^p F(m+n+x)([P]_m \times [Q]_n)$, where $i = 0$ for client and $i = 1$ for server, and secret shares $[P]_m, [Q]_n \in \mathbb{F}_p$ are multiplied using Beaver triple multiplication (Beaver, 1991). We depict the full preprocessing phase operation in Figure 3 and present the algorithmic details below.

---

**Algorithm 1** Tabula Preprocessing Phase

---

1: client $\longleftarrow$ random secret $s_0 \in \mathbb{F}_p$
2: server $\longleftarrow$ random secret $s_1 \in \mathbb{F}_p$
3: client, server locally initialize table $F' \in \mathbb{F}_p^{p \times p \times p}$ s.t. $F'[i][j][x] = F(i+j+x)$
4: client computes $P \in \mathbb{F}_p^p$ s.t $P[i] = \begin{cases} 1 & i = s_0 \\ 0 & x \neq s_0 \end{cases}$

5: server computes $Q \in \mathbb{F}_p^p$ s.t $Q[i] = \begin{cases} 1 & i = s_1 \\ 0 & x \neq s_1 \end{cases}$

6: client, server exchange shares of $P, Q$ to obtain secret shares $[P], [Q]$
7: client, server compute $[PQ] \in \mathbb{F}_p^{p \times p}$ where $[PQ][i][j] = [P][i] \times [Q][j]$ via Beaver triple multiplication
8: client, server compute $[T][x] = \sum_{m=0}^p \sum_{n=0}^p F'[m][n][x] \times [PQ][m][n]$ for all $i \in \mathbb{F}_p$

---

**Preprocessing Phase Correctness**
In this protocol the client and server specify the coordinate of $s$ through an outer product of their indicator vectors $P, Q$, which sets $[T][x] = [F(s+x)]$. This computes the correct answer as

$$[T][x] = \sum_{m=0}^p \sum_{n=0}^p F(m+n+x)([P]_m \times [Q]_n)$$

$$= \sum_{m=0}^p \sum_{n=0}^p F(m+n+x)([P_m \times Q_n]) \text{ (Beaver triple multiplication)}$$

$$= \sum_{m=0}^p \sum_{n=0}^p F(m+n+x) \times \begin{cases} [1] & m = s_0 \text{ and } n = s_1 \\ [0] & otherwise \end{cases}$$

$$= [F(s_0 + s_1 + x)]$$

$$= [F(s+x)]$$

**Preprocessing Phase Security**
Security and privacy is preserved as each step of the protocol consists entirely of secure steps: secret sharing $P$ and $Q$ leaks no information about the vectors (hence leaks no information about either $s_0$, $s_1$, and $s$), and Beaver triple multiplication is likewise secure (Beaver, 1991). Concretely, we see that the only communication between client and server occurs when they exchange blinded secrets (i.e: $[P], [Q]$) and when they perform Beaver triple multiplication. As these steps leak no information to either party about the underlying secrets, the client and server compute $[T]$ without leaking any information about $s_0, s_1$ and hence leak no information about $s$.

**Preprocessing Phase Communication and Computation Cost**
The bulk of the preprocessing phase lies in computing an outer product between $P, Q$. We perform this outer product just once and reuse it across $i, x$ in $T_i[x]$. Hence, the protocol requires performing just a single outer product between vectors $\in \mathbb{F}_p^p$. This incurs $O(p^2)$ Beaver triple multiplication operations, and assuming that a sufficient number of Beaver triples were generated before the preprocessing phase, communication cost is naively $O(p^2 \log(p))$ bits assuming that the values of the vectors are each $\log(p)$ bits.

This naive $O(p^2 \log(p))$ communication cost can be significantly reduced to $O(p^2)$ by having $P, Q$ be secret shared *binary* vectors, rather than be shared in $\mathbb{F}_p$, then doing a conversion back to $\mathbb{F}_p$ after the final inner product. This can be done as the true values of the vectors $P$, $Q$ are either 0 or 1. Concretely, upon reception of the binary shares of $P$ or $Q$ the current party computes

$$[T_i(x)] = \sum_{m=0}^{p} \sum_{n=0}^{p} F(m + n + x) * [PQ^T][m, n]$$

Observe that $[T_i(x)]_1 - [T_i(x)]_2$ is either $F(s + x)$ or $-F(s + x)$ (in the case that the first party has the 1 and the second party has the 0 in the selected index, and the reverse). We perform an extra Beaver triple multiplication by the correction factor to eliminate this potential negation (by multiplying it by the parity of the sum of $[PQ^T]$), which costs an extra $O(\log(p))$ bits of communication per inner-product. Since there are only $p$ inner products, these correction factors cost a negligible $O(p \log(p))$ communication. With this optimization, preprocessing communication cost is now $O(p^2)$ bits. As $p$, the quantized field size of the activation domain, is set to be extremely small (i.e.: less than or equal to 256 for 8-bit quantized activations), preprocessing communication costs $2(256)^2 = 2^{16} = 131072$ bits = 16 KB per table (the factor of two at the front is because Beaver triple multiplication requires both parties exchange secret shared values, and we have $256^2$ Beaver triple multiplication operations). This is comparable to the 17.5 KB cost that garbled circuits with full precision requires (Mishra et al., 2020a).

We emphasize that the prior analysis assumed that Beaver triples were obtained beforehand in a pre-preprocessing phase; we think this is reasonable that in a practical scenario parties would obtain sufficient amounts of Beaver triples for any protocol due to their importance. However, accounting for Beaver triple preprocessing, communication cost is still an asymptotic $O(p^2)$ bits assuming that Beaver triples were generated using oblivious transfer e.g: Nielsen et al. (2012), which requires just 2 OT calls to generate 1-bit Beaver triples. Using the OT procedure proposed in Huang et al. (2022) the concrete cost of a single OT is 3 bits for 1-bit values. Hence, pre-preprocessing costs for the Beaver triples would still be $O(p^2)$ bits, with a higher constant factor burden. On a concrete example of 8-bit activations, the cost for preprocessing the Beaver triples would amount to $6 \times 256^2$ bits = 48 KB of communication. While this exceeds the 17.5 KB cost of garbled circuits, we believe that the online benefits of TABULA more than make up for this detriment.

In terms of computation, we see that for precomputing a single table, we require summing across $p^2$ values (each entry of the outer product) for every entry of the table. Since there are $p$ entries in the table, computation costs scale as $O(p^3)$. However, these operations may be efficiently vectorized and parallelized as they are standard matrix operations. For 8-bit tables, this is around 16 million field operations.

### 3.4 Note on Tabula Security

Although the TABULA protocol assumes a semi-honest threat model as inherited from the Delphi (Mishra et al., 2020a) framework, more generally TABULA's online phase which consists of utilizing a secure lookup procedure is information-theoretically secure (Ishai et al.). This is intuitive as all communication between parties are randomly blinded by an additive factor. This is another advantage that TABULA holds over garbled circuits implementations many of which are only computationally secure up to a security parameter in the semi-honest setting (mpc).

## 4 Results

We present results showing the benefits of TABULA over garbled circuits for secure neural network inference. We evaluate our method on neural networks including a large variant of LeNet for MNIST, ResNet-32 for Cifar10, and ResNet-34 / VGG-16 for Cifar-100, which are relatively large image recognition neural networks that prior secure inference works benchmark (Ghodsi et al., 2021; Mishra et al., 2020a; Jha et al., 2021; van der Hagen & Lucia, 2021). Unless otherwise stated, we compare against an implementation of the Delphi protocol (Mishra et al., 2020a) using garbled circuits for nonlinear activation functions, without neural architecture changes, during the online inference phase. As before, we use 64-bit fields for TABULA to reduce the impact of the secure truncation protocol, however, for the baseline implementation that uses garbled circuits we use

32-bit fields to reduce communication cost. Experiments are run on AWS c5.4x large machines (US-West1 (N. California) and US-West2 (Oregon)) which have 8 physical Intel Xeon Platinum @ 3 GHz CPUs and 32 GiB RAM; network bandwidth between these two machines achieves a maximum of 5-10 Gbit/sec, according to AWS. We use the same machine/region specs as detailed in (Mishra et al., 2020a), but with 2x more cores/memory (c5.4xlarge vs c5.2xlarge).

To ensure fair comparison, we compare Tabula against garbled circuits with quantized inputs, specifically garbled circuits with 32-bit, 16-bit, and 8-bit inputs, which are commonly used precisions, but we also show more detailed results on a more granular level by fixing accuracy/precision and comparing systems costs between our methods. With the same activation precision, both Tabula and garbled circuits compute the same result. Like other works we benchmark using a batch size of 1 (Mishra et al., 2020a; Rathee et al., 2020; Huang et al., 2022; Ghodsi et al., 2021). Tabula is feasible only for precision up to 12-bits due to the exponential storage costs required for each extra bit of precision, hence, we show results up until this limit. Garbled circuits on the other hand can obtain 32/16-bit precision, however this is beyond the range of precision that Tabula may handle to avoid large storage costs. Hence, we may show results for 16/32-bit garbled circuits as a baseline reference, but the main comparison is between 8-bit garbled circuits and 8-bit Tabula, or between the two approaches when obtaining a fixed accuracy.

## 4.1 Communication Reduction

**ReLU Communication Reduction**
We benchmark the amount of communication required to perform a single ReLU with Tabula vs garbled circuits. Table 2 shows the amount of communication required by both protocols during online inference. Tabula achieves significant ($> 280\times$) communication reduction compared to garbled circuits. Note our implementation of garbled circuits on 32-bit inputs achieves the same communication cost as reported by (Mishra et al., 2020a) (2KB communication for 32-bit integers).

| Garbled Circuits (32-bit) | Garbled Circuits (16-bit) | Garbled Circuits (8-bit) | Tabula | Comm. Reduction (vs 32/16/8 bit GC) | | |
|---|---|---|---|---|---|---|
| 2.17KB | 1.1KB | .562KB | 2B | $1112\times$ | $560\times$ | $280\times$ |

Table 2: Tabula with 8-bit activations vs garbled circuits communication cost for one ReLU.

We additionally compare ReLU communication of our protocol against recent works like CrypTflow2 (Rathee et al., 2020) and Cheetah (Huang et al., 2022). CrypTflow2 and Cheetah similarly utilize a tree-based secure comparison protocol dependent on oblivious transfer (Rathee et al., 2020; Huang et al., 2022). However unlike CrypTflow2, Cheetah swaps out the underlying oblivious transfer implementation for a more efficient version Huang et al. (2022). Our following analysis assumes that CrypTFlow2 uses a more efficient OT protocol based on preprocessing which reduces the online communication costs beyond what they present in their paper; broadly, the tree-based comparison method that CrypTflow2 utilizes requires at least 6 calls to 1-out-of-128 oblivious transfer for optimal communication complexity (Rathee et al., 2020), which, with preprocessing, takes at least $6 \times 128 = 768$ bits or 96 bytes, as oblivious transfer with preprocessing requires sending all $n$ bits to the original sender at the end (Beaver, 1995; Naor & Pinkas, 1999). Tabula requires just 16 bits of communication regardless of the nonlinear function being computed, obtaining a $48\times$ improvement in communication over the tree based comparison method of CrypTflow2 / Cheetah assuming the use of this preprocessing-based OT method. Cheetah's approach on the other hand uses the same tree-based comparison approach (Huang et al., 2022) but swaps out the underlying OT method for a more efficient version; specifically, Cheetah's communication cost is $11 \times L$ where $L$ is the bitlength of the field element, which results in 88 bits of communication for 8-bit values and 352 bits for 32-bit values. Tabula requires just 16 bits of communication, which represents a $5.5\times$ and $22\times$ reduction respectively. SiRNN (Rathee et al., 2021), another paper which utilizes an OT based protocol, uses more communication than that of ReLU of CrypTflow (Rathee et al., 2020), hence our method would see $> 48\times$ communication improvement when compared to their approach. We also compare communication cost against FSS approaches (Boyle et al., 2019; Gupta et al., 2022; Agarwal et al., 2022; Ryffel et al., 2021). Generally, for the ReLU op Tabula obtains the

same 2B communication cost as FSS, however TABULA obtains several notable qualitative advantages over FSS, and a table comparison is shown in Table 3. We summarize these communication cost comparisons in Table 4, which compares the online communication cost of a single ReLU operation for TABULA, CryptFlow2 and Cheetah.

| | **Tabula** | **FSS** |
|---|---|---|
| Computational Efficiency | 8-bit memory access per op | >= 1 PRG (i.e: AES-128) operation per op |
| Generality | 2B comm. cost for any nonlinear function (must fit in table) | Comm. cost increases for more complex nonlinear ops |
| Security | Information-Theoretically Secure | Computational Security (up to security parameters $\lambda$) |
| Complexity | Table lookup | DPF / DCF (Boyle et al., 2020) |

Table 3: Qualitative comparison between TABULA and FSS schemes.

| **Tabula** | **CrypTFlow2 (OT preproc. method)** | **Cheetah 32-bit** | **Cheetah 8-bit** | **FSS (8-bit; ReLU op)** | **Comm. Reduction** |
|---|---|---|---|---|---|
| 2B | 96B | 44B | 11B | 2B | $48 \times$ / $22 \times$ / $5.5 \times$ / $1 \times$ |

Table 4: TABULA (8-bit activations) vs CrypTFlow2 (Rathee et al., 2020) and Cheetah (Huang et al., 2022) and FSS (Boyle et al., 2019; 2020) online communication cost for performing a single ReLU operation during the online phase. For CrypTFlow2 the communication is based on an OT method that uses preprocessing which achieves better online communication cost than what is described in Rathee et al. (2020). Note FSS, Cheetah, CrypTFlow2 costs are specific to the ReLU op, while TABULA communication cost is the same for any function provided they are quantized down to a sufficiently small table size.

**Total Online Communication Reduction**
We benchmark the total amount of online communication required during the online phase of a single private inference for various network architectures including LeNet, Resnet-32, ResNet-34 and VGG (batch size 1). Table 5 shows the number of ReLUs per network, as well as the communication costs of using garbled circuits (for 32/16/8 bit inputs) vs TABULA. TABULA reduces communication significantly ($> 20\times$, $> 10\times$, $> 5\times$ vs 32,16,8 bit garbled circuits) across various network architectures.

| Network | ReLUs | Garbled Circuits (32-bit) | Garbled Circuits (16-bit) | Garbled Circuits (8-bit) | Tabula | Comm. Reduction (vs 32/16/8 bit GC) | | |
|---|---|---|---|---|---|---|---|---|
| LeNet | 58K | 124 MB | 62 MB | 31 MB | 3.5 MB | $35.4\times$ | $17.7\times$ | $8.8\times$ |
| ResNet-32 | 303K | 311 MB | 155 MB | 77 MB | 14 MB | $22.2\times$ | $11.1\times$ | $5.6\times$ |
| VGG-16 | 276K | 286 MB | 143 MB | 72 MB | 12.1 MB | $23.6\times$ | $11.8\times$ | $5.6\times$ |
| ResNet-34 | 1.47M | 1.5 GB | .75 GB | 370 MB | 59.5 MB | $24.7\times$ | $12.4\times$ | $6.2\times$ |

Table 5: TABULA vs garbled circuits total online communication cost during secure inference for different network architectures.

We additionally compare end-to-end communication costs against Rathee et al. (2020), the current state-of-the-art for neural network inference, on various networks Minionn and ResNet34(Liu et al., 2017), shown in Table 6. TABULA's compact tables enable much lower communication costs during the online phase of secure neural network inference, leading to an order of magnitude reduction in communication costs.

Finally, Figure 4 shows the communication reduction TABULA achieves compared to garbled circuits with $A_n$-bit quantized inputs at a fixed accuracy threshold, and shows TABULA achieves over $8-9\times$ communication reduction across networks to maintain close to full precision accuracy. These values reflect total online

| Network | ReLUs | Tabula | CrypTFlow2 | Comm. Reduction |
|---|---|---|---|---|
| MinioNN | 176K | 25MB | 280MB | 11.2 × |
| ResNet-34 | 1.47M | 59.5 MB | 590MB | 9.9 × |

Table 6: Tabula vs CrypTFlow2 (Rathee et al., 2020) end-to-end communication cost for performing secure neural network inference on selected networks (Minionn (Liu et al., 2017) CIFAR10 architecture, and ResNet34 CIFAR100).

communication costs, not just ReLU communication costs, and hence we find we are primarily bottlenecked by the communication for the linear layers rather than nonlinear layers. Also, we do not make any architectural changes to the neural network (e.g: replace any ReLU operations with quadratic operations, retrain, etc).

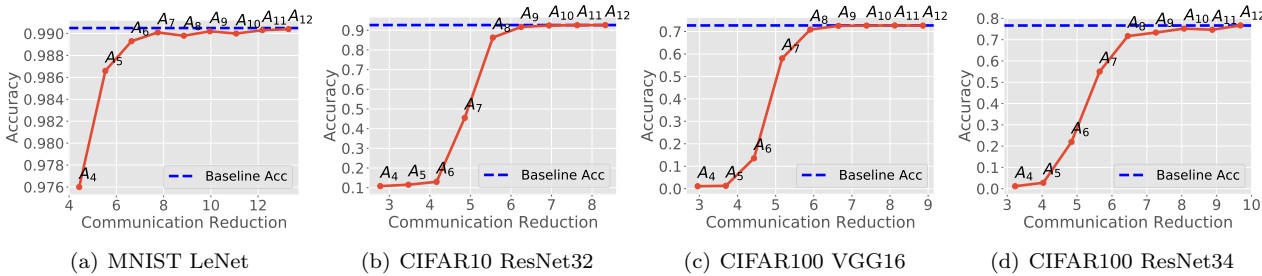

(a) MNIST LeNet  (b) CIFAR10 ResNet32  (c) CIFAR100 VGG16  (d) CIFAR100 ResNet34

Figure 4: Tabula communication reduction improvement over garbled circuits (x-axis) vs accuracy, when garbled circuits communication cost for $A_n$ bits is compared against Tabula with 8-bit inputs. Tabula with 8-bit inputs achieves accuracy equivalent to $A_8$. This plot shows Tabula with 8-bit inputs obtains greater communication reduction over GCs with different precisions on various test architectures, while remaining withing 1% to 2% of the full precision baseline. Because both Tabula and GC see linear scaling of communication cost with the number of activation bits to the input of the nonlinear function, $n$-bit Tabula vs $n$-bit GC would yield a communication reduction factor of 6-8× for all $n$ on these test architectures. Also the labels denoted $A_n$ mark the accuracy achieved by both Tabula and GCs should they use $n$ bit activations, as both compute the same result. Baseline precision shown as the dashed blue line.

## 4.2 Storage Costs

We compare the storage savings Tabula achieves against garbled circuits. Recall that Tabula requires storing a single lookup table for each nonlinear activation call. This storage cost grows exponentially with the size of its tables, which dictates the precision of the activations. Using less storage means reducing the precision for the activations of the neural network and introduces some amount of error into the nonlinear function. This creates a tradeoff between storage and network accuracy. Similar to garbled circuits, Tabula tables must be stored on both client and server, and likewise, storage costs are equivalent for both client and server; hence, in our results we show the storage cost for a single party. Below we show both the storage savings for a single ReLU operation disregarding the accuracy impact from the quantization, and additionally the storage vs accuracy tradeoffs for various networks (LeNet, ResNet32/34, VGG). Storage costs directly translate to memory usage costs at inference time since the lookup tables or garbled circuits must be loaded into memory to be used to evaluate the nonlinear functions.

**ReLU Storage Savings vs Precision**
We compare the storage use between Tabula and garbled circuits for a single ReLU operation. Tabula's storage use is the the size of its table multiplied by the number of bits of elements in the original field, which we default to 64-bit numbers. Garbled circuits, on the other hand, uses 17KB, 8.5KB, and 4.25KB for each 32-bit, 16-bit, and 8-bit ReLU operation respectively (Mishra et al., 2020a).

Figure 5 presents the storage usage of both Tabula and garbled circuits for a single ReLU operation, and shows that Tabula achieves comparable storage use to garbled circuits at precisions 8-10, and lower storage

use with precisions below 8. Specifically, with 8 bits of precision for activation TABULA achieves an 8.25×, 4.1× and 2× savings vs 32-bit, 16-bit and 8-bit garbled circuits; with ultra low precision TABULA achieves even more gains (4 bits yields around 136× storage reduction vs 32-bit garbled circuits and 17× reduction vs 8-bit garbled circuits). These results imply that standard techniques to quantize activations down below 8 bits and advanced techniques to quantize below 4 bits (Ni et al., 2020; de Bruin et al., 2020; Zhao et al., 2020) can be applied with TABULA to achieve significant storage savings. Notably, TABULA achieves storage savings at ultra low precision activations as a 1-bit reduction in activation precision yields a 2× storage reduction, unlike for garbled circuits where storage is reduced linearly.

**Storage Savings and Accuracy Tradeoff**

We present TABULA's total storage usage versus accuracy tradeoff in Figure 6. In this experiment, we directly quantize the network's activations during execution time uniformly across layers, recording the achieved accuracy and memory/storage requirements for a single inference. As shown in Figure 6, across various tasks and network architectures, activations may be quantized to 9 bits or below while maintaining within 1-3% accuracy. This allows TABULA to achieve comparable or even less storage use than garbled circuits at a fixed accuracy threshold. We emphasize that future work may apply more advanced quantization techniques (Ni et al., 2020; de Bruin et al., 2020) to reduce activation precision below 8-bits and achieve even better storage savings. Our results here show that

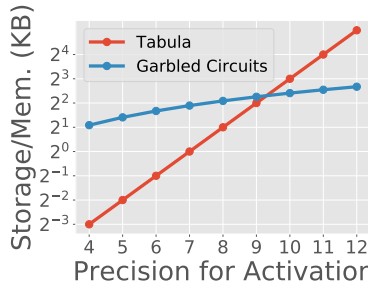

Figure 5: TABULA and garbled circuits storage use for a single ReLU operation.

even with very basic quantization techniques, TABULA achieves comparable storage usage versus garbled circuits, and indicate that TABULA is more storage efficient as fewer bits of precision for the activations are used.

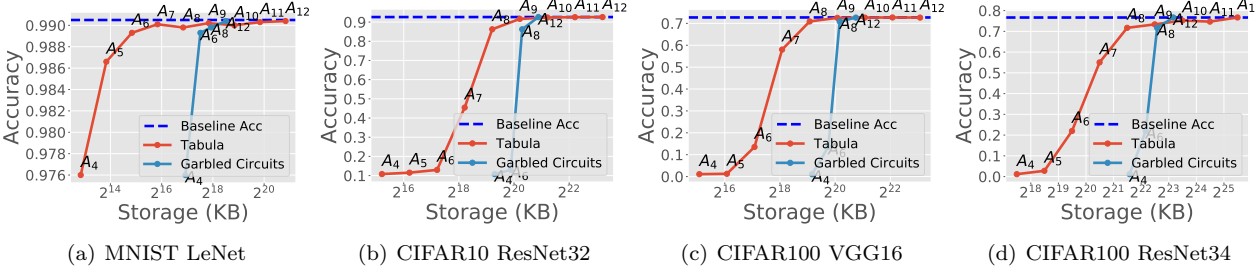

Figure 6: TABULA overall storage usage for a single inference versus accuracy for different tasks and neural networks. Each point is annotated with $A_n$, specifying the precision of activations for that run. With activation precisions above 10 TABULA uses more storage than garbled circuits due to the exponential increase in the size of its tables; however, below a precision of 8, TABULA achieves notable storage savings ($> 2\times$) over garbled circuits. Baseline precision shown as the dashed blue line.

### 4.3 Runtime Speedup

We compare the runtime speedup TABULA achieves over garbled circuits. As noted in various secure neural network inference works (Mishra et al., 2020a; Ghodsi et al., 2021; Cho et al., 2021), executing nonlinear activation functions via garbled circuits takes up the majority of secure neural network execution time, hence, replacing garbled circuits with an efficient alternative has a major impact on runtime. Below we present the TABULA's runtime benefits when executing individual ReLU operations and when executing relatively large state-of-the-art neural networks.

**ReLU Runtime Speedup**

Table 7 shows the runtime speedup TABULA achieves over garbled circuits when executing a single ReLU

| Garbled Circuits 32-bit Runtime (us) | Garbled Circuits 16-bit Runtime (us) | Garbled Circuits 8-bit Runtime (us) | Tabula Runtime (us) | Tabula Speedup (vs 32/16/8 bit GC) | | |
|---|---|---|---|---|---|---|
| 184 | 111 | 69 | .55 | 334 × | 202 × | 105 × |

Table 7: Tabula runtime speedup vs garbled circuits on a single ReLU operation. Tabula is orders of magnitude faster than garbled circuits.

operation. Tabula achieves over 100× runtime speedup due to its simplicity: the cost of transferring 16 bytes of data and a single access to RAM is orders of magnitude faster than garbled circuits. Our implementation of garbled circuits on 32-bit inputs is slower than reported in Delphi (Mishra et al., 2020a). Our implementation of garbled circuits takes around 184 us per ReLU, whereas the reported is 84 us (Mishra et al., 2020a). However, even if the implementation in Delphi achieves an optimal 4× speedup with 8-bit quantization, Tabula is still 38× faster.

**Neural Network Runtime Speedup**
We present Tabula's overall speedup gains over garbled circuits across various neural networks including LeNet, ResNet32/34 and VGG16. Table 8 and Figure 7 shows that Tabula reduces runtime by up to 50× across different neural networks, bringing execution time below 1 second per inference for the majority of the networks. Bigger networks are increasingly bottlenecked by ReLU operations, and hence Tabula's runtime reduction increases in magnitude with the size of the neural network under consideration. Figure 8 shows a breakdown of where execution time is being spent, for both Tabula and garbled circuits. As shown, Tabula reduces the runtime spent on computing activation functions by up to orders of magnitudes. With bigger networks, the impact of executing nonlinear activation functions is larger. Hence, Tabula sees greater runtime improvement on larger networks. Additionally, in the runtime breakdown chart, the linear layers for Tabula were considerably slower than the linear layers when using garbled circuits – we believe that cache effects caused this difference in performance, as Tabula keeps all tables in memory, which may have overflowed to swap memory. Despite the slowdown in the linear layers, this has negligible impact on runtime due to non-linear layers being the dominant cost, and Tabula sees considerably performance gains by being faster on the nonlinear layers.

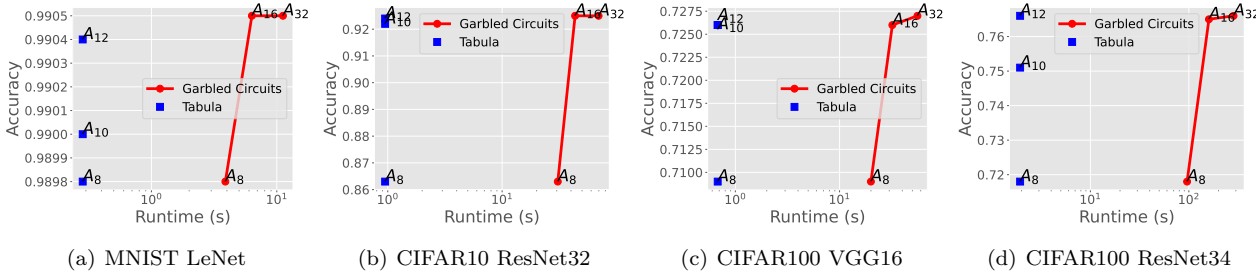

(a) MNIST LeNet     (b) CIFAR10 ResNet32     (c) CIFAR100 VGG16     (d) CIFAR100 ResNet34

Figure 7: Tabula overall runtime for a single inference versus accuracy for different tasks and neural networks. Each point is annotated with $A_n$, specifying the precision of activations for that run. At activation precisions 10-12 (achieving within 1-2% of baseline accuracy), Tabula achieves significant runtime speedup ($> 10\times$) over garbled circuits.

## 4.4 Preprocessing Costs

We benchmark our proposed algorithm for preprocessing Tabula tables against garbled circuits preprocessing times to demonstrate that Tabula preprocessing costs are comparable to garbled circuits.

**Preprocessing Runtime & Communication Costs**

| Network | ReLUs | 32-bit Garbled Circuits Runtime (s) | 16-bit Garbled Circuits Runtime (s) | 8-bit Garbled Circuits Runtime (s) | Tabula Runtime (s) | Speedup (vs 32/16/8 bit GC) | | |
|---|---|---|---|---|---|---|---|---|
| LeNet | 58K | 11.1 | 6.3 | 3.9 | .29 | 38.3× | 21.7× | 13.4× |
| ResNet-32 | 303K | 69.7 | 43.4 | 30.6 | .97 | 71.8× | 44.7× | 31.5× |
| VGG-16 | 284.7K | 55.9 | 32.1 | 19.9 | .67 | 83.4× | 47.9× | 29.7× |
| ResNet-34 | 1.47M | 284.3 | 159.9 | 95.9 | 1.85 | 153.7× | 86.4× | 51.8× |

Table 8: TABULA total online runtime speedup compared with garbled circuits. Compared to garbled circuits, TABULA achieves significant runtime speedup during neural network execution by reducing code complexity, communication costs, and memory/storage overheads.

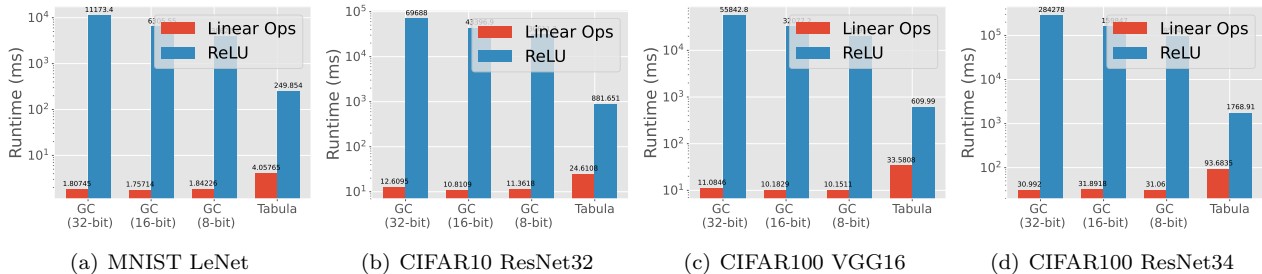

(a) MNIST LeNet    (b) CIFAR10 ResNet32    (c) CIFAR100 VGG16    (d) CIFAR100 ResNet34

Figure 8: Runtime breakdown across linear and nonlinear (ReLU) layers comparing TABULA with 8-bit inputs and garbled circuits with 32,16,and 8-bit inputs. TABULA achieves significant performance gains on nonlinear layers, leading to major runtime speedups.

We compare runtime and communication costs for initializing a single ReLU operation. Table 9 shows the cost of preprocessing a single ReLU operation for Tabula with 8-bit inputs, and garbled circuits. In terms of communication costs, Tabula is comparable to GC with 32-bit inputs; however, Tabula requires more communication than GC with 16/8 bit inputs. In terms of runtime, Tabula generally requires significantly more computation than garbled circuits, leading to higher runtime. The majority of Tabula preprocessing runtime is spent towards computing field operations for performing the multiply-add-accumulate operation between the outer product and the nonlinear function (recall that computation costs for an 8-bit input scales as $O(256^3)$). These computation costs can be significantly decreased through further parallelization and vectorization.

| Metric | Tabula Preprocessing (8-bit) | Garbled Circuits Preprocessing (32-bit) | Garbled Circuits Preprocessing (16-bit) | Garbled Circuits Preprocessing (8-bit) |
|---|---|---|---|---|
| Runtime (ms) | 6 | .155 | .092 | .053 |
| Communication (b) | 16384 | 17920 | 8960 | 4480 |

Table 9: Tabula vs Garbled Circuits runtime and communication preprocessing costs for a single ReLU operation. Note: Tabula preprocessing costs, like runtime costs, stay constant regardless of activation function, unlike garbled circuits.

We further show the effect of number of bits used for the activation function on preprocessing communication costs. As each bit that is eliminated reduces the size of the table by a factor of 2, saving a single bit exponentially decreases runtime and communication costs. As seen in Figure 9, at around 5 bits TABULA preprocessing communication costs become lower than communication cost for garbled circuit at the same bitwidth.

## 4.5 End-to-end Preprocessing Communication Costs

We additionally compare end-to-end preprocessing communication costs across various models (LetNet, ResNet, VGG) between TABULA and Garbled Circuits. Table 10 shows a comparison of the communication costs between different models. Again, TABULA requires more communication than GC with 8/16-bit inputs but less than GC with 32-bit inputs, due to the need for computing outer products using Beaver triples that scale with the cardinality of the field. Although this is costly, results show that preprocessing can be feasibly performed at similar cost to GC with 32-bit inputs. Further research and algorithmic developments may drive down the preprocessing cost of initializing TABULA tables.

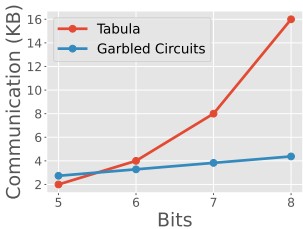

Figure 9: Tabula preprocessing communication cost vs Garbled Circuits for different number of bits.

| Network | ReLUs | Tabula Preprocessing (8-bit) | Garbled Circuits Preprocessing (32-bit) | Garbled Circuits Preprocessing (16-bit) | Garbled Circuits Preprocessing (8-bit) |
|---|---|---|---|---|---|
| LeNet | 58K | 906 MB | 991 MB | 496 MB | 248 MB |
| ResNet-32 | 303K | 4.6 GB | 5.05 GB | 2.53 GB | 1.27 GB |
| VGG-16 | 284.7K | 4.3 GB | 4.75 GB | 2.37 GB | 1.19 GB |
| ResNet-34 | 1.47M | 22.4 GB | 24.5 GB | 12.25 GB | 6.13 GB |

Table 10: Preprocessing communication cost comparison between TABULA and garbled circuits for various neural network models. TABULA has comparable preprocessing costs compared to garbled circuits.

## 5 Conclusion

TABULA is a secure and efficient protocol for computing nonlinear activation functions in secure neural network inference. Our approach obtains considerable computational benefits over garbled circuits and other approaches to securely computing nonlinear functions. To conclude, we point out the following observation: quantization, as applied to improve standard neural network performance, typically obtains sublinear runtime improvements (as low bitwidth ops typically do not scale linearly in perf. with bits due to hardware inefficiencies), and linear memory improvements. Through our method, quantization as applied to secure neural network inference, obtains super-linear runtime/communication improvements that scale with the complexity of the underlying nonlinear operation, and exponential memory improvements. We believe that, quantization, an already important performance improvement technique for neural networks, will be even more crucial for secure neural network inference, and that our method TABULA is a key approach towards realizing this fact. Additionally, TABULA will see improvement to both the online and offline phases with further advancements to neural network quantization. TABULA is a step towards sustained, low latency, low energy, low bandwidth real time secure inference applications.

### Acknowledgements

Michael Mitzenmacher was supported in part by NSF grants CCF-2101140, CNS-2107078, and DMS-2023528.

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

## A  Appendix

**Polynomial approximation vs quantization**
Ample research has been dedicated towards exploring how to make polynomial approximations more amenable to neural networks (as enabling polynomial activations eliminates system bottlenecks imposed by nonlinear functions). However a significant body of evidence demonstrates that polynomial activations face remarkable barriers to achieving high accuracy, especially for deep networks; on the other hand, research indicates that large and deep neural networks (including networks like VGG, ResNet, LSTMs and transformers on tasks like Cifar100 and ImageNet) may be quantized to 8 bits and below (oftentimes to 4 or even 2 or 1 bit) with little loss of accuracy. Hence, rather than use polynomial activations, our work suggests that quantization is the more preferred approach. This is a significant observation driving our approach. Below, we show a comparison of the accuracy performance of quantization vs polynomial activation.

| Method | Baseline | Polynomial | 2-bit Activations |
|---|---|---|---|
| Accuracy | 93.29% | 71.81% | 91.5% |

Table 11: Comparison of polynomial activations vs quantization on ResNet32 Cifar10.

| Method | Baseline | Polynomial | 3-bit Activations |
|---|---|---|---|
| Accuracy | 74.39% | 65.17% | 73% |

Table 12: Comparison of polynomial activations vs quantization on ResNet18 Cifar100.

Tables 11 and 12 compares the final test accuracy achieved by polynomial activations vs quantization for ResNet32 Cifar10 and ResNet18 Cifar100 respectively. These results were obtained from Garimella et al. (2021); Choi et al. (2019); Hoang et al. (2020). As seen, polynomial approximations significantly harm accuracy, while activation quantization, even at very low precision (2-bit / 3-bit) results in near lossless accuracy. This phenomenon extends to large datasets such as ImageNet where 4-bit ResNet50 achieves lossless performance Abdolrashidi et al. (2021), whereas polynomial activations incur significant accuracy loss on tiny imagenet Garimella et al. (2021). In many of these quantization works, accuracy performance includes the quantization of the weights as well as the activations; as our approach requires only activation quantization, it may be inferred that even better accuracies may be attained than what these numbers indicate.

In the experiments shown in the main text, we do not do quantized retraining as in the results immediately above (and instead directly apply quantization to pretrained weights; this is known as post-training quantization), hence, there is room for accuracy improvement over what was demonstrated in the main text. This reaffirms the potential of our approach in achieving efficient and accurate secure neural network inference.

**Note on truncation errors**
All accuracies reported in our paper are obtained by running the protocols in full and account for truncation error resulting from the secure truncation protocol. Notably, in our implementation, we maintain a single separate static scale parameter per layer that is known to both client and server (leaking negligible model information), ensuring that the underlying integer values of secretly shared activations are maintained between $[0, 2^{14}]$ (this is a common technique when performing quantized inference with limited bitwidth datatypes for acceleration on hardware). As P(catastrophic truncation error) is proportional to the chance that the secret blinding factor is less than the secret value, the probability of catastrophic error for one truncation is $2^{14-64}$. This means, there is a 99.98% chance that all 1,000,000 calls to a network with 300,000 ReLUs succeeds. With just 80 bits, P(catastrophic error) is $2^{14-80}$, leading to a 99.99959% prob. that 1,000,000,000 calls to a 300,000 ReLU network all succeed. The common off by one errors, like quantization error, has negligible impact on model quality (a similar finding by Huang et al. (2022)). This is a key detail and difference from prior works (which do not maintain a separate scale, significantly inflating catastrophic truncation error probability). Detecting truncation error and retrying them is a topic for future work.

