# OpenReview forum: "Tabula: Efficiently Computing Nonlinear Activation Functions for Secure Neural Network Inference"
_TMLR — Accepted by TMLR_

### Review · Reviewer_qKEu · 2024-01-03

**Summary Of Contributions:**

Tabula offers a new approach to enhancing the efficiency of secure neural network inference, presenting a secure lookup table algorithm as an alternative to garbled circuits. The algorithm is designed to address the communication, storage, and runtime challenges associated with traditional garbled circuits, aiming to provide a more practical solution. Through the use of precomputed lookup tables and quantization techniques, Tabula modestly achieves reductions in communication while offering improved end-to-end inference speed for neural networks compared to existing protocols.

**Audience:**

Yes

**Claims And Evidence:**

Yes

**Requested Changes:**

Overall, I think the paper is well-written, and introduces a new technique to solve an important problem. I see that in that the authors have already addressed some of the writing clarity issues in the previous iterations. Thus, I am happy with the paper as it is. (Please note that I'm not an expert on this topic.)

**Strengths And Weaknesses:**

Strengths:
- Under given circumstances (with 8-bit quantized activations) Tabula achieves several cost benefits by preprocessing tables for computing the activation functions
- The theoretical analysis and rationale behind Tabula is well-written
- The evaluation comparison is comprehensive and the limitations are explicitly stated

Weaknesses:
- The approach is limited to 8-bit quantization or lower. 16/32-bit representation would require huge memory

The paper is well written and as an outsider to this topic, I could follow most of the algorithms. Except for the following part:
- How are vectors P and Q shared between the client and server without revealing $s_0$ and $s_1$?

---

> ### Author Response · Authors · 2024-03-12
> **Response to Reviewer qKEu**
>
> Thank you for the constructive feedback!
>
> ---
>
> “Approach is limited to 8-bit quantization or lower, 16/32-bit representation would require huge memory”
>
>
> This is indeed a drawback of our method, however, we highlight that 8-bit quantization is a standard and widely adopted optimization applied to neural networks and has been demonstrated to have negligible impact on network accuracy[1], and recent research shows that 8-bit activations can even be applied to LLMs as shown in this recent work[2] which states “[BitNet] is trained from scratch, with 1.58-bit weights and 8-bit activations”. We emphasize that the general trend of applying quantization (specifically activation quantization) to models to optimize them for performance reinforces the viability of our approach.
>
> [1] NVIDIA Int8 Quantization - https://developer.nvidia.com/blog/achieving-fp32-accuracy-for-int8-inference-using-quantization-aware-training-with-tensorrt/
>
> [2] The Era of 1-bit LLMs: All Large Language Models are in 1.58 Bits - https://arxiv.org/abs/2402.17764
>
> ---
>
> “How are vectors P and Q shared … without revealing s_0 and s_1?”
>
> s_0 and s_1 determine the index that P and Q respectively set to 1 (everywhere else is 0). However, when constructing the lookup table, the two parties only interact with secret shares of P and Q, denoted as [P], [Q] in the figures and paper. So P and Q are additively blinded with a random vector, and hence s_0 and s_1 are not leaked, since no information about P and Q are leaked during the protocol.
>
> ---

---

### Review · Reviewer_BgbH · 2024-01-12

**Summary Of Contributions:**

The paper proposes Tabula, a novel protocol for efficiently computing nonlinear activation functions for secure neural network inference, based on secure lookup tables and quantization. The authors describe a secure and practical algorithm for initializing Tabula tables in the offline preprocessing phase, and analyzes its security, communication, and storage properties. Empirical evaluation on various standard neural networks and datasets shows that Tabula achieves significant improvements at inference time over existing methods based on garbled circuits or function secret sharing in terms of communication, runtime, and storage costs, while maintaining comparable accuracy.

**Audience:**

Yes

**Claims And Evidence:**

Yes

**Requested Changes:**

In addition to the writing changes suggested above, I had the following questions:

1. At the beginning of Section-4 you mention that Tabula is feasible only for precision up to 12 bits but you also say that you use 64-bit fields for Tabula. Can you clarify the difference between precision and field size in this context?

2. Why is the communication cost of Tabula fixed at 16 bytes regardless of the precision of activation inputs?

**Strengths And Weaknesses:**

Strengths:

1. Based on the survey of prior work it seems that even though lookup tables have demonstrated speedup over garbled circuits in other applications this is the first work to successfully applying them to secure neural network inference by leveraging the fact that neural network activations can be quantized without much loss in accuracy, and this can save memory and make it feasible to construct a lookup table for the entire space.

2. The paper demonstrates that their approach, Tabula, achieves significant improvements over garbled circuits and other state-of-the-art methods in terms of communication, runtime, and storage costs, while maintaining comparable accuracy and security.

Weaknesses:

1. The evaluation shows that there is a significant overhead in terms of both runtime and communication cost at the preprocessing stage for Tabula as compared to the garbled circuits approach. However, I appreciate the fact that the authors have acknowledged this and I believe it is okay to leave the optimization of this stage for future work.

2. The description of the approach is hard to follow due to the long paragraphs and in-line equations in Section 3. I would strongly recommend revising the section, splitting the long paragraphs into multiple short paragraphs and avoiding the use of in-line equations as far as possible. It may also be worth including a description of garbled circuits in the appendix for interested readers who are unfamiliar with them.

---

> ### Author Response · Authors · 2024-03-12
> **Response to Reviewer BgbH**
>
> Thank you for the constructive feedback!
>
> ---
>
> “Significant overhead in terms of both runtime and communication cost at the preprocessing stage for Tabula”
>
> On this point, we would like to emphasize that neural networks may be quantized to below 8-bits for the activations as demonstrated in some notable prior research on quantization[1]. This would exponentially reduce the runtime and communication cost of the preprocessing stage. Further advancements in research towards lower precision neural networks would benefit our lookup table approach.
>
> [1] Post-training 4-bit quantization of convolution networks for rapid-deployment - https://arxiv.org/abs/1810.05723
>
> ---
>
> “Long paragraphs and in-line equations in Section 3 … splitting the long paragraphs into multiple short paragraphs”
>
> We have updated section 3 and made it more readable by splitting the paragraphs and by pulling out major inlined equations.
>
> ---
>
> “Tabula is feasible only for precision up to 12 bits but you also say that you use 64-bit fields for Tabula. Can you clarify the difference between precision and field size in this context”
>
> 64-bit fields are used to represent the secret shares of the fixed-point values that are used as operands to the matrix-multiplication (i.e: linear operations)	, whereas the 8/12-bits are for the activations. Our protocol is equivalent to quantizing a 64-bit value down to 8-bits, applying the nonlinear function, then recasting these values back to 64-bit numbers, then continuing to do a linear operation over them (in other words, the linear operations are actually done in 64-bit precision, it is just the activation function inputs that are quantized). The number of entries of the lookup table is 2^12 (in case of 12-bit activations), but the number of bits of each element of the lookup table can be any number of bits, though in our implementation to optimize storage we use 8-bit quantized entries, then scale back up to 64-bit for the subsequent linear computation. We revise to emphasize this distinction in Section 3.2 “Secure Truncation”.
>
> ---
>
> “Why is the communication cost of Tabula fixed at 16 bytes regardless of the precision of activation inputs?”
>
> We miscommunicated this point and have clarified this caption in Figure 4. We meant to say that Tabula’s communication cost is independent of the complexity of the nonlinear function being computed. We have updated Figure 4 which made this incorrect statement. The results of Figure 4 have not changed, but we have updated the caption to improve the description of these results.

---

### Review · Reviewer_iJ91 · 2024-03-04

**Summary Of Contributions:**

This work introduces TABULA an algorithm for computing nonlinear activation functions efficiently for multiparty secure neural network inference. This paper claims that unlike traditional approaches that rely on garbled circuits for secure inference of non linear activations, TABULA is efficient in terms of communication, storage and runtime. It operates within the Delphi framework and focuses on improving the costs associated with non linear activations. The approach relies on secure lookup from precomputed tables and is done in two phases - offline phase where lookup tables are computed on both server and client side and online phase wherein the non linear activation is performed via table lookup. The lookup table is shared across two parties and contains output to every possible input to that non-linear function. Quantization is used throughout to avoid exponential growth in storage costs. During  online phase the inputs are securely truncated to match the quantized range in the lookup tables. The tables and inputs are secret shared between server and client to avoid either party being aware of the complete input/model weights.

Experiments are presented to evaluate the algorithm and compared to the quantized version of garbled circuits. Results show that TABULA has reduced communication/storage/runtime costs without much degradation in accuracy.

**Audience:**

Yes

**Broader Impact Concerns:**

No Concern

**Claims And Evidence:**

Yes

**Requested Changes:**

See Weakness

**Strengths And Weaknesses:**

**Strengths**:
- The paper is well written - the motivation, problem formulation, algorithm and results are all explained well.
- The paper employs a novel combination of lookup tables/function secret sharing/quantization to make the online phase efficient
- Extensive experimentation is done (some is missing, see weakness) to indicate the advantages of TABULA in terms of cost savings - communication and runtime majorly.

**Weaknesses**
- Baseline methods: TABULA relies on quantization so it makes sense to compare with quantized versions of baseline. But results on comparing the accuracy with and without quantization should also be presented. Accuracy comparison is only presented for 8bit TABULA and garbled circuits. Since TABULA relies on quantization, the reader should know how much drop in accuracy to expect because of that.
- Feasibility of the method for only up-to 12 bits is a major drawback. I think it should be mentioned in the abstract/introduction to set the readers expectations.
- Regarding storage costs, the claim that TABULA uses comparable storage is not very apparent from the experiments. In a practical setting (say up-to 12 bits) TABULA uses way more storage than garbled circuits, in an 8 bit setting it uses 2x the storage.
- Are there any networks other than image classification that can be explored for experimentation?
- Some recent works have emerged using lookup tables as well, authors might like to compare/mention it in their work https://eprint.iacr.org/2024/369

---

> ### Author Response · Authors · 2024-03-12
> **Response to Reviewer iJ91**
>
> Thank you for the constructive feedback!
>
> ---
>
> “Results on comparing the accuracy with and without quantization should also be presented”
>
> We highlight that we have these results presented in Figure 4 and Figure 6 (the points labeled A_n show the accuracy drops for different activation precisions for Tabula), and shows that quantizing down to 8-bits still achieves accuracy close to the full-precision baseline.
>
> ---
>
> “Feasibility of the method for only up-to 12 bits is a major drawback.”
>
> We agree that this is a drawback and will highlight this in our abstract + body. We furthermore emphasize that quantization is a widely adopted standard in deploying neural networks to the real world and refer to our response to reviewer qKEu in our justification of the use of quantization.
>
> ---
>
> “the claim that TABULA uses comparable storage is not very apparent from the experiments. In a practical setting (say up-to 12 bits) TABULA uses way more storage than garbled circuits, in an 8 bit setting it uses 2x the storage”
>
> We would like to clarify that our claim is made with regards to comparing to Tabula with 8-bit activations, which in this case is only 2x more storage. We have updated the abstract to clarify these points.
>
> ---
>
> “Are there any networks other than image classification that can be explored for experimentation?”
>
> Indeed we see application to other types of networks (i.e: language, LSTM) as potential future work, especially on non-ReLU nonlinear operations, where our approach would likely see greater gains since our approach’s cost does not depend on the complexity of the nonlinear function. In our current work, we focus primarily on image recognition tasks, like in many private neural network inference works[1,2,3].
>
> [1] Delphi - https://www.usenix.org/conference/usenixsecurity20/presentation/mishra
>
> [2] DeepReduce - https://arxiv.org/abs/2103.01396
>
> [3] CrypTFlow2 - https://eprint.iacr.org/2020/1002.pdf
>
> ---
>
> “Some recent works have emerged using lookup tables as well, authors might like to compare/mention it in their work”
>
> We have added the referred work to our related works discussion. We note that our paper was written before the work that the reviewer listed had appeared and are happy to include it.

---

### Author Response · Authors · 2024-03-12
**Response to Reviewers**

We thank the reviewers for their thorough and constructive feedback. We report a log of changes made in response to the reviewer’s feedback below and also responded to the questions and concerns of each individual reviewer below their reviews. We have uploaded a new version of the paper with changes highlighted in red. We are happy to make further changes upon request and answer followup questions.

# Changes
* Made Section 3 more readable by splitting paragraphs and pulling out major inlined equation (Section 3)
* Update Section 3.2 secure truncation to note the distinction between field size and activation bits (Section 3.2)
* Update Figure 4 to clarify the accuracy vs communication reduction plot (Figure 4)
* Update abstract to clarify that Tabula uses 8-bit quantization + comparable = within factor of 2 (Abstract)
* Update related works with note on lookup table approach for GCs (Related Works)

---

> ### Author Response · Authors · 2024-03-12
> **Update**
>
> Update: Updated Figure 4 caption + Section 3.2 again to improve clarity

---

### Decision · Action_Editor_5s6H · 2024-06-08

**Recommendation:** Accept as is

**Comment:**

This work introduces an algorithm for computing nonlinear activation functions for multiparty secure neural network inference. The proposed method is well-motivated and the authors have conducted solid comparison to demonstrate the benefit of the method. Reviewers raised several questions about the experiments which are well addressed in the revision. Therefore all the reviewers and AC think the paper is ready to publish in TMLR.

**Audience:**

The paper is proposing a novel method for secure neural network inference, which will be interested to people working on security & AI. The proposed algorithm is novel and we all think the audience working on the area of secure neural network inference will be interested in this paper.

**Claims And Evidence:**

The claims made in the submission are convincing and well supported by experiments.